# Comparison between single and serial computed tomography images in classification of acute appendicitis, acute right-sided diverticulitis, and normal appendix using EfficientNet

So Hyun Park[1☯], Young Jae Kim[2☯], Kwang Gi Kim[2‡*], Jun-Won Chung[3‡], Hyun Cheol Kim[4], In Young Choi[5], Myung-Won You[6], Gi Pyo Lee[7], Jung Han Hwang[1]

1 Department of Radiology, Gil Medical Center, Gachon University College of Medicine, Incheon, South Korea, 2 Department of Biomedical Engineering, Gachon University, Gil Medical Center, Incheon, South Korea, 3 Division of Gastroenterology, Department of Internal Medicine, Gil Medical Center, Gachon University College of Medicine, Incheon, South Korea, 4 Department of Radiology, Kyung Hee University Hospital at Gangdong, Seoul, South Korea, 5 Department of Radiology, Korea University Ansan Hospital, Ansan, South Korea, 6 Department of Radiology, Kyung Hee University Hospital, Seoul, South Korea, 7 Department of Health Sciences and Technology, Gachon Advanced Institute for Health Sciences and Technology (GAIHST), Gachon University, Incheon, South Korea

☯ These authors contributed equally to this work.
‡ KGK and JWC also contributed equally to this work.
* kimkg@gachon.ac.kr

**Data Availability Statement:** All relevant data are within the paper and its Supporting Information files.

## Abstract

This study aimed to develop a convolutional neural network (CNN) using the EfficientNet algorithm for the automated classification of acute appendicitis, acute diverticulitis, and normal appendix and to evaluate its diagnostic performance. We retrospectively enrolled 715 patients who underwent contrast-enhanced abdominopelvic computed tomography (CT). Of these, 246 patients had acute appendicitis, 254 had acute diverticulitis, and 215 had normal appendix. Training, validation, and test data were obtained from 4,078 CT images (1,959 acute appendicitis, 823 acute diverticulitis, and 1,296 normal appendix cases) using both single and serial (RGB [red, green, blue]) image methods. We augmented the training dataset to avoid training disturbances caused by unbalanced CT datasets. For classification of the normal appendix, the RGB serial image method showed a slightly higher sensitivity (89.66 vs. 87.89%; $p = 0.244$), accuracy (93.62% vs. 92.35%), and specificity (95.47% vs. 94.43%) than did the single image method. For the classification of acute diverticulitis, the RGB serial image method also yielded a slightly higher sensitivity (83.35 vs. 80.44%; $p = 0.019$), accuracy (93.48% vs. 92.15%), and specificity (96.04% vs. 95.12%) than the single image method. Moreover, the mean areas under the receiver operating characteristic curve (AUCs) were significantly higher for acute appendicitis (0.951 vs. 0.937; $p < 0.0001$), acute diverticulitis (0.972 vs. 0.963; $p = 0.0025$), and normal appendix (0.979 vs. 0.972; $p = 0.0101$) with the RGB serial image method than those obtained by the single method for each condition. Thus, acute appendicitis, acute diverticulitis, and normal

**Funding:** This research was supported by the Ministry of Science and ICT (MSIT), Korea, under the Information Technology Research Center (ITRC) support program (IITP−2021−2017−0−01630) super-vised by the Institute for Information and communications Technology Promotion (IITP), the GRRC program of the Gyeonggi province (No. GRRC-Gachon2020 (B01), and the research was supported by G−ABC FRD2019−11−02(3).

**Competing interests:** The authors have declared that no competing interests exist.

**Abbreviations:** CNN, convolutional neural network; CT, computed tomography; RGB, red, green, blue; ROI, region-of-interest.

appendix could be accurately distinguished on CT images by our model, particularly when using the RGB serial image method.

## Introduction

Acute lower abdominal pain is a common symptom among patients visiting hospitals [1, 2]. Abdominopelvic computed tomography (CT) is commonly used to evaluate acute right lower abdominal pain in adults with suspected acute appendicitis or acute right-sided diverticulitis [1, 3, 4]. Although radiologists can easily distinguish between these conditions on CT images, the rapidly increasing volume of CT examinations in the emergency room can place a burden on radiologists [5]. Distinguishing between these diseases and a normal appendix is also sometimes difficult for a physician (e.g., an internal medicine specialist or a surgeon). Although clinical presentations of the two diseases are similar, the treatments differ between surgical and medical treatments [4, 6–8], with acute appendicitis requiring surgery and acute diverticulitis requiring medical treatment in uncomplicated cases [9, 10]. Thus, correct classification of each condition in patients with acute abdominal pain is crucial for rapid and accurate decision-making regarding appropriate treatment [8].

A convolutional neural network (CNN) is a deep learning-based neural network and can be used to analyze radiological image data. A CNN was recently used to diagnose acute appendicitis automatically in a single-center, single-protocol cohort of 319 pre-contrast abdominopelvic CT images [11]. However, to the best of our knowledge, CNNs using contrast-enhanced abdominopelvic CT images to classify acute appendicitis, acute diverticulitis, and normal appendix have not been developed to date. Although CNNs using single-image analysis can outperform conventional machine-learning methods [12–14], they fail to leverage the depth information from CT images (e.g., the tubular structure of the appendix on serial axial CT images as opposed to the round structure of the appendix on a single cross-sectional axial CT image). Consecutive CT image slices with red, green, and blue (RGB) channel superposition provide information that can be used to improve the automated diagnostic performance, and, in cases of serial CT imaging, implement deep learning methods based on CNNs [15, 16].

The EfficientNet algorithm consists of compounded coefficients for scaling the depth, width, and resolution dimensions of a CNN [17]. EfficientNet, which presents a new approach for performing transfer learning in classification, reduces the analysis time and computing resources required, and consequently, its use in various deep learning studies has increased recently [18–21]. This algorithm is composed of 8 models, from B0 to B7, with increasing numbers referring to more parameters and higher accuracy [17, 22]. This influences the neural architecture used to explore the baseline. EfficientNet-B0 has a better trade-off on accuracy and calculation time (i.e., floating-point operations per second) [17, 23]. We have previously used the B0 model. which covers 4 million trainable parameters [24].

In the present study, we used a large dataset of CT images to develop and validate a CNN, using the EfficientNet algorithm and the corresponding training method, for the automated classification of acute appendicitis, acute diverticulitis, and normal appendix. We also compared the diagnostic performance of the CNN when applied to single CT images versus when applied to serial CT images with the RGB superposition method.

## Methods

### Study population

Approval for this retrospective study was obtained from the institutional review board (GDIRB2020-096) of the Gil medical center. The requirement for obtaining written informed

patient consent was waived because of the retrospective nature of the study. The study was conducted according to the principles of the Declaration of Helsinki.

We searched the CT database and electronic medical records of patients who visited a hospital complaining of acute lower abdominal pain and who underwent contrast-enhanced abdominopelvic CT between January 2017 and August 2019 and were diagnosed with acute appendicitis or right-sided diverticulitis. For the acute appendicitis group, the inclusion criteria were as follows: (a) patients ($\geq$ 20-years-old) with acute lower abdominal pain who were diagnosed with acute appendicitis that was detected via CT (i.e., we searched CT reports to find patients with acute appendicitis and reviewed their clinical symptoms) and surgically confirmed, and (b) who had undergone preoperative abdominopelvic CT within 1 week of surgery. The exclusion criteria were as follows: (a) cecal cancer with secondary appendicitis or appendiceal cancer with appendicitis and (b) severe motion artifacts on CT images. For the acute right-sided diverticulitis group, the inclusion criteria were as follows: (a) patients ($\geq$ 20-years-old) with acute lower abdominal pain and acute right-sided diverticulitis detected via CT, who were (b) medically treated and improved during follow-up ($>$ 1 month). The exclusion criteria were (a) bowel perforation due to acute diverticulitis and (b) severe motion artifacts on CT images. For the normal appendix group, we identified all patients aged $\geq$ 20 years and searched CT reports between January 2017 and December 2020 to find those with a normal appendix by using the key words: "normal appendix" (n = 410). The exclusion criteria were (a) CT images showing acute diverticulitis or colitis and (b) severe motion artifacts on CT images. Among these 410 cases, we randomly selected 215 patients, to ensure a similar number of CT slides (n = 1296) as for the acute appendicitis (n = 1959) and acute diverticulitis (n = 823) cases (Fig 1). Selected patients were retrospectively enrolled in the study, and their data were collected in our imaging laboratory.

## CT imaging

The CT scanners used in this study included SOMATOM Edge, SOMATOM Definition AS, SOMATOM Definition Flash, and SOMATOM Force (Siemens Healthcare, Erlangen, Germany). The scan parameters varied: tube voltages of 80, 100, or 120 kVp and reference tube currents of 170–298 mAs. No low-dose CT protocol was used for these datasets. The CT images were reconstructed on the axial plane with a thickness of 2–5 mm.

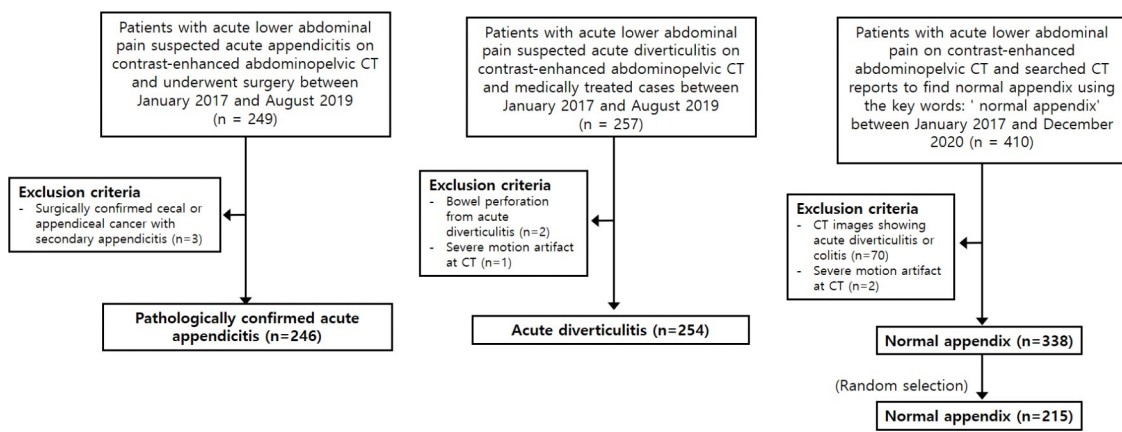

**Fig 1. Study flowchart.**

## Validation of the deep learning model

We used five-fold cross-validation and a test dataset to train the deep learning model. For each fold, the dataset samples were randomly assigned as follows: 60–70% to a training dataset to train and optimize the model; 15–20% to a validation dataset to evaluate the best configuration and tune the model parameters; and 15–25% to a test dataset to evaluate its classification performance. In addition, the training dataset was analyzed after the augmentation of acute appendicitis, diverticulitis, and normal appendix data to prevent training disturbances caused by data imbalances. More details on model training are provided in the Implementation and training of the deep learning model subsection of the methods section.

## Preprocessing with region-of-interest extraction and RGB method superposition

We manually cropped box-shaped regions-of-interest (ROIs) at the same position across all axial planes from the start to the end positions of the visible targets for acute appendicitis, acute diverticulitis, and normal appendix. The cropped images were then resized to 224 × 224 pixels based on an image size conversion reference [25]. Next, we obtained three consecutive images from the CT slice sequence (sorted along the *z*-position) and represented them in the RGB method to acquire a new color image. Fig 2 illustrates the process for obtaining a CT image with the RGB superposition method.

## Implementation and training of the deep learning model

We used the IBM Power System AC922 server with four Tesla V100s, NVLink GPUs, and 16 GB memory (NVIDIA Corp., Sta. Clara, CA, USA) to run the deep learning model based on the EfficientNet-B0 architecture (Fig 3A). The code was implemented in Python 3.7.11 and TensorFlow frameworks (Version 2.2.0) were run on the Ubuntu 18.04.3 operating system.

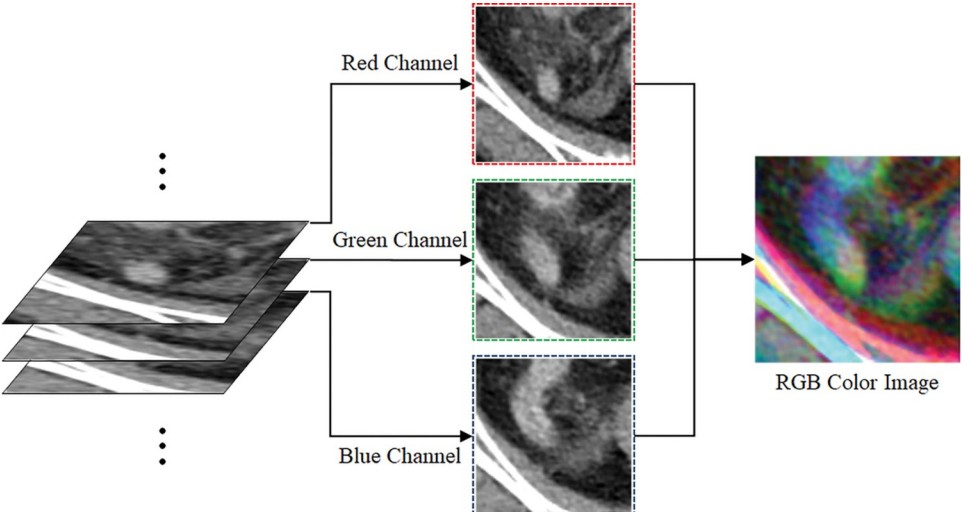

**Fig 2. Process for obtaining serial computed tomography (CT) images with red, green, blue (RGB) channel superposition.** The RGB superposition method enhances the differences in the connectivity and shape of the target in serial CT images. The connected region of a continuous slice is almost gray in color, whereas the unconnected region (s) show(s) different shapes (i.e., primary colors). A three-dimensional effect can be obtained from two-dimensional images based on the representation of the connectivity information pertaining to the organization that can be confirmed by the three-dimensional volume using the RGB color model.

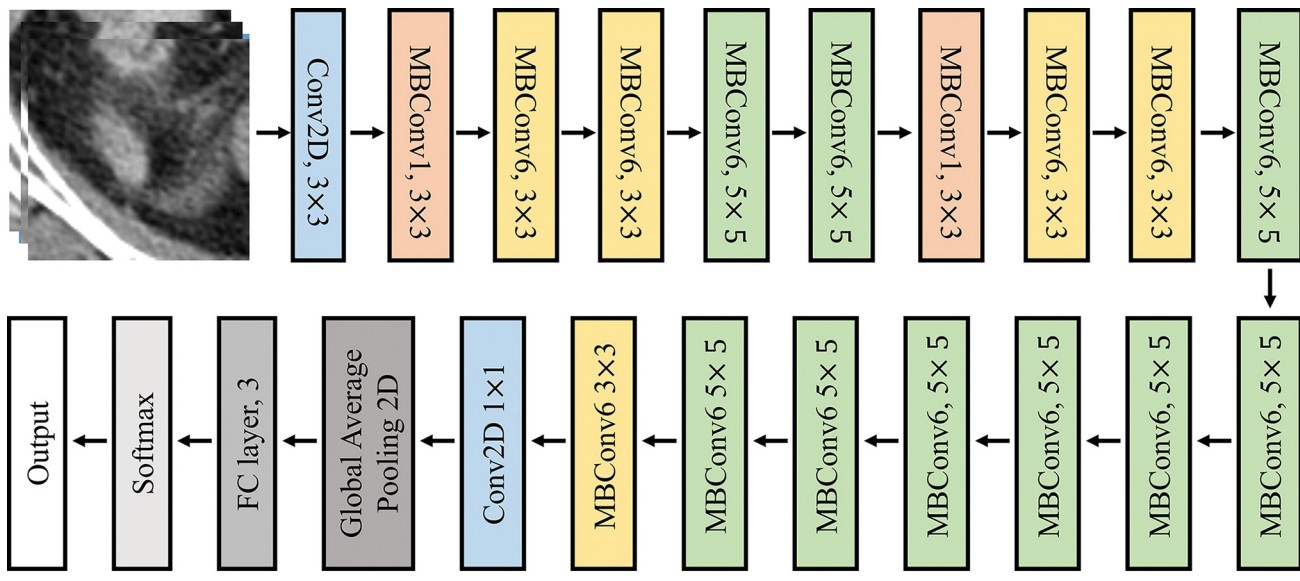

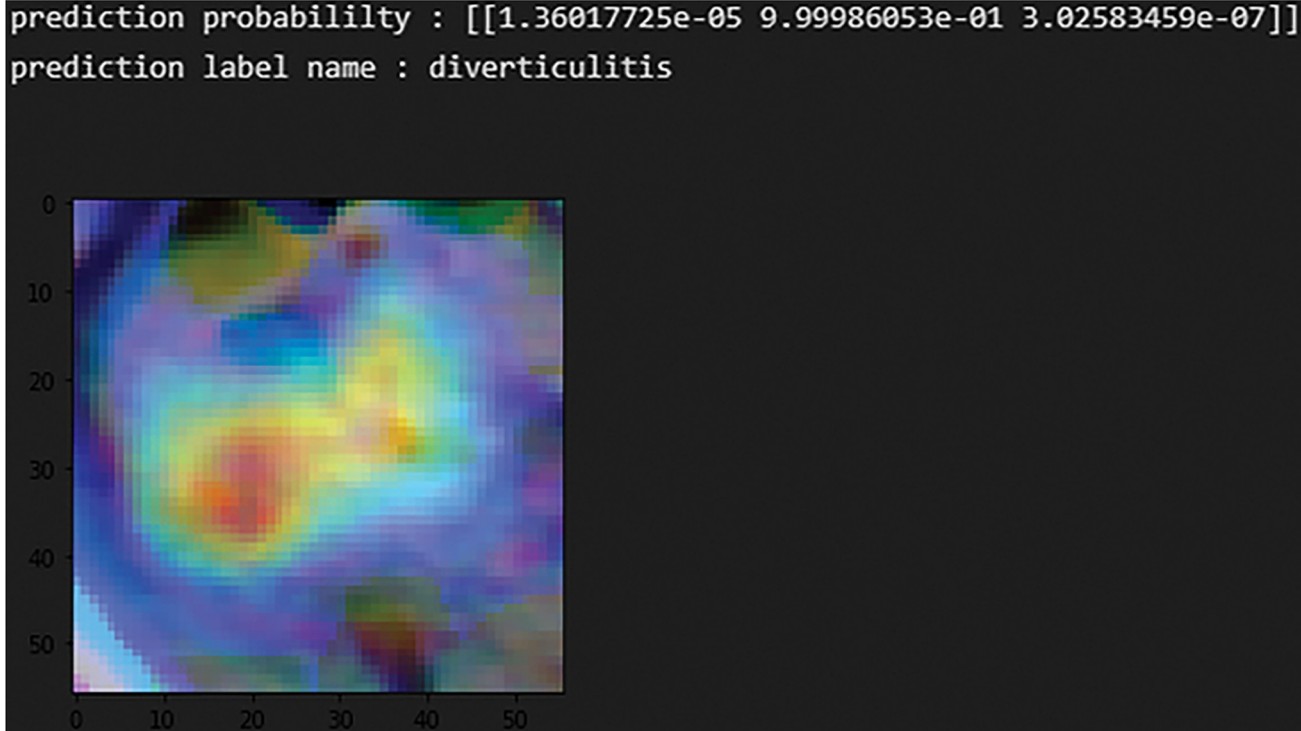

**Fig 3.** **(a)** EfficientNet-B0 architecture **(b)** Example of a prediction outcome label.

The code for image pre-processing was implemented on OpenCV-Python 3.4.10, Pydicom 2.2.2 and read-roi 1.6.0.

For training, we set the batch size to 64, number of epochs to 200, and used Adam as the optimizer [26]. We adapted early stopping to avoid overfitting during training; thus, the training was stopped when the validation loss did not improve after over 10 epochs. We described the loss function using categorical cross-entropy [27]. In addition, we set the learning rate to

0.0001 (1e-4), and for cases with more than 30 epochs, the learning rate was set to 0.00001 (1e −5) to ensure that this rate does not increase significantly as learning progresses. Application of the deep learning model yielded the prediction probability for the three statuses (probability for acute appendicitis, acute diverticulitis, or normal appendix). Among the three, the status with the highest probability was taken as the prediction label (Fig 3B).

## Statistical analysis

The diagnostic accuracy, sensitivity, and specificity of CNN classification of acute appendicitis, acute diverticulitis, and normal appendix using the single image and serial image RGB methods were calculated and determined by considering 95% confidence intervals. The CNN classification performances of the two methods were compared in terms of the area under the curve (AUC) values, obtained from receiver operating characteristic curves, and sensitivity, using the McNemar test. $P < 0.05$ was considered as statistically significant. The statistical analyses were conducted using the MedCalc software (version 18.11.3) and IBM SPSS Statistics for Windows (version 21.0; IBM Corp., Armonk, NY, USA).

## Results

We analyzed the training, test, and validation datasets using the data obtained from 715 consecutive patients (347 women, 368 men; age, 44.3 ± 18.4 years). Of these, 246 had acute appendicitis, 254 had acute diverticulitis, and 215 had normal appendix. The details of the test dataset are presented in Table 1. The test dataset contained 4,078 CT image slices (acute appendicitis, 1,959; acute diverticulitis, 823; normal appendix, 1,296), which were derived from the scans of 715 patients. More details on the model case and the data obtained using five-fold cross-validation are presented in Table 2 and S1 Appendix. As can be seen, the amount of training data (number of CT slides) among the three groups after data augmentation are relatively similar.

The diagnostic performance of the two CNN classification approaches are listed in Table 3. For the classification of acute appendicitis, the RGB method demonstrated slightly higher sensitivity (87.85 vs. 85.60%; $p = 0.047$), accuracy (87.94% vs. 86.07%), and specificity (88.01% vs. 96.04%) than the single image method. For the classification of acute diverticulitis, the RGB method yielded slightly higher sensitivity (83.35 vs. 80.44%; $p = 0.019$), accuracy (93.48% vs. 92.15%), and specificity (96.04% vs. 95.12%) than the single image method. For the classification of normal appendix, the RGB method showed slightly higher sensitivity (89.66 vs. 87.89%; $p = 0.244$), accuracy (93.62% vs. 92.35%), and specificity (95.47% vs. 94.43%) than the single method. For classification of the three statuses, the RBG method revealed a higher overall accuracy than the single method (87.51% vs. 85.29%). The RGB method showed a significantly higher mean AUC for acute appendicitis (0.951 vs. 0.937; $p < 0.0001$), acute diverticulitis (0.972 vs. 0.963; $p = 0.0025$), and normal appendix (0.979 vs. 0.972; $p = 0.0101$) than the single image method (Fig 4).

Detailed comparisons between the single image and RBG methods in the classification of the three statuses are presented in Table 4 and Figs 5–7. Among the 1,959 CT images of acute appendicitis, 146 and 123 images were misclassified as acute diverticulitis and 136 and 115

**Table 1. Characteristics of the datasets constructed in this study.**

| Variable | Total | Acute appendicitis | Acute diverticulitis | Normal appendix |
|---|---|---|---|---|
| No. patients | 715 | 246 | 254 | 215 |
| No. data (CT image slices) | 4078 | 1959 | 823 | 1296 |
| Age (years) | 44.3 ± 18.4 | 41.9 ± 19.2 | 44.6 ± 13.6 | 46.7 ± 21.9 |
| Women:men (men, %) | 347:368 (51.5) | 116:130 (52.8) | 117:137 (56.0) | 114: 101 (47.0) |

**Table 2. Train, test, and validation case and data.**

| cv | name | Train case* | Test case | Train data∫ | Validation data | Test data | Total case | Total data |
|----|------|-------------|-----------|-------------|-----------------|-----------|------------|------------|
| cv1 | Appendicitis | 199 | 47 | 2506∬ (1253)^ | 314 | 392 | 246 | 1959 |
| | Diverticulitis | 198 | 56 | 2630 (526) | 132 | 165 | 254 | 823 |
| | Normal appendix | 171 | 44 | 2472 (824) | 207 | 265 | 215 | 1296 |
| cv2 | Appendicitis | 192 | 54 | 2484 (1242) | 311 | 406 | 246 | 1959 |
| | Diverticulitis | 213 | 41 | 2619 (524) | 131 | 168 | 254 | 823 |
| | Normal appendix | 172 | 43 | 2466 (822) | 206 | 268 | 215 | 1296 |
| cv3 | Appendicitis | 195 | 51 | 2486 (1243) | 311 | 405 | 246 | 1959 |
| | Diverticulitis | 199 | 55 | 2605 (521) | 131 | 171 | 254 | 823 |
| | Normal appendix | 169 | 46 | 2435 (812) | 204 | 280 | 215 | 1296 |
| cv4 | Appendicitis | 195 | 51 | 2492 (1246) | 312 | 401 | 246 | 1959 |
| | Diverticulitis | 204 | 50 | 2615 (523) | 131 | 169 | 254 | 823 |
| | Normal appendix | 170 | 45 | 2460 (820) | 206 | 270 | 215 | 1296 |
| cv5 | Appendicitis | 203 | 43 | 2566 (1283) | 321 | 355 | 246 | 1959 |
| | Diverticulitis | 202 | 52 | 2690 (538) | 135 | 150 | 254 | 823 |
| | Normal appendix | 178 | 37 | 2598 (866) | 217 | 213 | 215 | 1296 |

Note: cv = cross validation

*Number of cases indicates the number of patients.

∫Number of data indicates the number of CT slides.

^Original number of CT slides and

∬augmentation number of CT slides.

images were predicted to be normal appendices by the single image and RBG methods, respectively. In Among the 823 CT images of acute diverticulitis, 142 and 126 images were predicted to be acute appendicitis and 19 and 11 images were predicted to be normal appendices using these two methods, respectively. Among the 1,296 CT images of normal appendices, 144 and 128 images were misclassified as representing acute appendicitis using the two methods, respectively.

## Discussion

We developed and applied a CNN model using the EfficientNet-B0 architecture to classify acute appendicitis, acute right-sided diverticulitis, and normal appendix from single and serial

**Table 3. Diagnostic performance of CNN classification using single and RGB methods for acute appendicitis and acute diverticulitis.**

| | Sensitivity | Specificity | Precision(PPV) | Accuracy | AUC | *p*-value* |
|---|-------------|-------------|----------------|----------|-----|-----------|
| Single | | | | | | |
| Appendicitis | 85.60(83.97–87.13) | 86.50(84.97–87.93) | 85.43(84.02–86.74) | 86.07(84.97–87.12) | 0.937 | <0.0001 |
| Diverticulitis | 80.44(77.56–83.10) | 95.12(94.32–95.83) | 80.63(78.09–82.94) | 92.15(91.28–92.96) | 0.963 | 0.0025 |
| Normal appendix | 87.89(85.98–89.61) | 94.43(93.51–95.25) | 88.02(86.30–89.56) | 92.35(91.49–93.15) | 0.972 | 0.0101 |
| RGB | | | | | | |
| Appendicitis | 87.85(86.21–89.27) | 88.01(86.55–89.37) | 87.14(85.78–88.39) | 87.94(86.90–88.92) | 0.951 | |
| Diverticulitis | 83.35(80.63–85.84) | 96.04(95.31–96.68) | 84.17(81.75–86.33) | 93.48(92.68–94.22) | 0.972 | |
| Normal appendix | 89.66(87.87–91.27) | 95.47(94.63–96.21) | 90.22(88.59–91.63) | 93.62(92.83–94.35) | 0.979 | |

Note.Data are represented as%, except for the AUC. Numbers in parentheses represent 95% confidence intervals.

*P-value was compared with the AUC values of the two methods.

## - (a) Single methods -

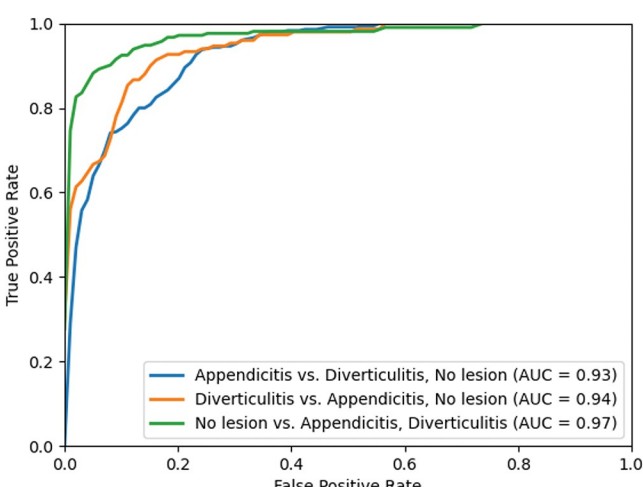

## - (b) RGB methods -

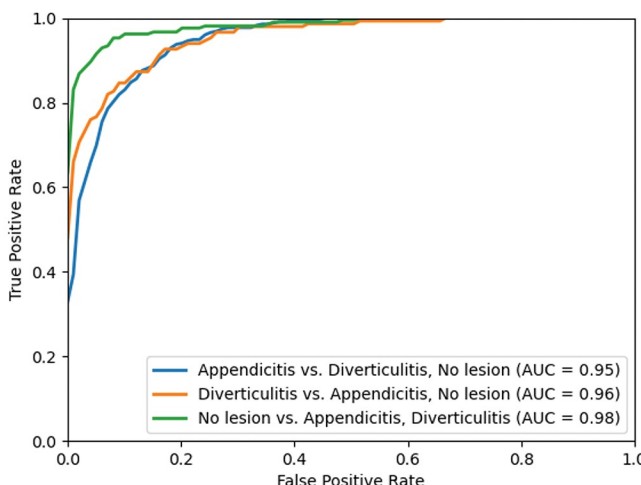

**Fig 4.** Area under the receiver operating characteristic curve for classifying acute appendicitis (blue), acute diverticulitis (orange), and normal appendix (green) using (a) the single image and (b) serial image red, green, blue (RGB) methods.

RGB contrast-enhanced abdominopelvic CT images. All the AUC values obtained using the RGB method were higher than those obtained using the single image method.

The sensitivity (87.89, 89.66%; single, RGB) and specificity (94.43, 95.47%; single, RGB) values for normal appendix were the highest among the three diseases. Appropriate classification of a normal appendix using a CNN could help inexperienced physicians identify a normal appendix from CT images in an emergency room. To improve the performance of our CNN,

**Table 4. Differences between the single and RGB methods for the data analyses.**

| Label | | | Single | | | Total data |
|---|---|---|---|---|---|---|
| | | | Predicted A | Predicted D | Predicted N | |
| Appendicitis | RGB | Predicted A | 1577 | 79 | 65 | 1721 |
| | | Predicted D | 56 | 62 | 5 | 123 |
| | | Predicted N | 44 | 5 | 66 | 115 |
| | Total data | | 1677 | 146 | 136 | 1959 |
| Diverticulitis | RGB | Predicted A | 82 | 37 | 7 | 126 |
| | | Predicted D | 59 | 621 | 6 | 686 |
| | | Predicted N | 1 | 4 | 6 | 11 |
| | Total data | | 142 | 662 | 19 | 823 |
| Normal appendix | RGB | Predicted A | 35 | 1 | 92 | 128 |
| | | Predicted D | 1 | 0 | 5 | 6 |
| | | Predicted N | 108 | 12 | 1042 | 1162 |
| | Total data | | 144 | 13 | 1139 | 1296 |
| Overall | RGB | Predicted A | 1694 | 117 | 164 | 1975 |
| | | Predicted D | 116 | 683 | 16 | 815 |
| | | Predicted N | 153 | 21 | 1114 | 1288 |
| | Total data | | 1963 | 821 | 1294 | 4078 |

Note.—Predicted A, predicted D, and predicted N refer to acute appendicitis, acute diverticulitis, and normal appendix predicted using each method.

## - (a) Single methods -

## - (b) RGB methods -

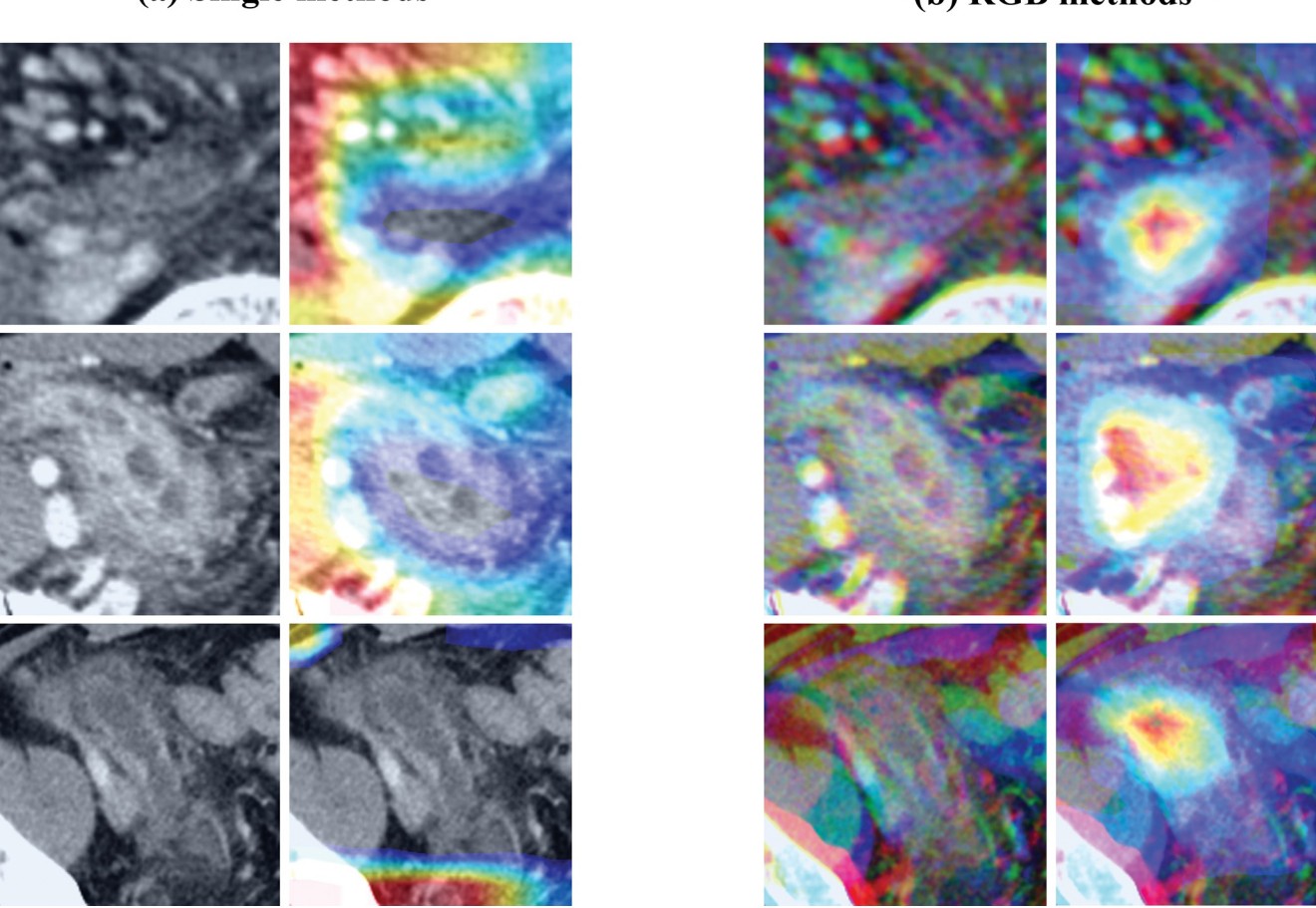

**Fig 5.** Examples of acute appendicitis cases correctly classified by the convoluted neural network by (CNN) using the (a) single image and (b) red, green, blue (RGB) methods, with and without class activation maps (CAMs).

we applied data augmentation on the training dataset and used the EfficientNet-B0 architecture. Although we enrolled a similar number of patients in each of the three groups, the amount of CT data in these groups varied significantly. For example, the appendix dataset involved a larger number of CT images than the acute diverticulitis dataset, because the appendix has a tubular structure, as can be seen from the several images on the axial CT scan, whereas diverticulitis has a small, round shape. For this reason, we applied data augmentation on the training dataset (particularly on the diverticulitis dataset) to minimize data imbalance to ensure that the training effectiveness is not hampered.

Using a region of low entropy and a fully convolutional network, Walid et al. [11] designed a CNN that could detect acute appendicitis on non-contrast CT images with a specificity of 68.8–92.6%. In the classification of acute appendicitis, our CNN was comparably more specific (86.50–88.01%). Unlike the model developed by Walid et al. [11], ours was developed using contrast-enhanced CT scans from a larger cohort. In practice, contrast-enhanced CT is more commonly used than non-enhanced CT in patients with suspected acute appendicitis. Our results showed that the developed model can be applied to clinical conditions when physicians need to classify acute appendicitis, normal appendix, and diverticulitis. Rajpurkar et al. [28] recently developed a CNN using video pre-training to classify acute appendicitis on contrast-

## - (a) Single methods -

## - (b) RGB methods -

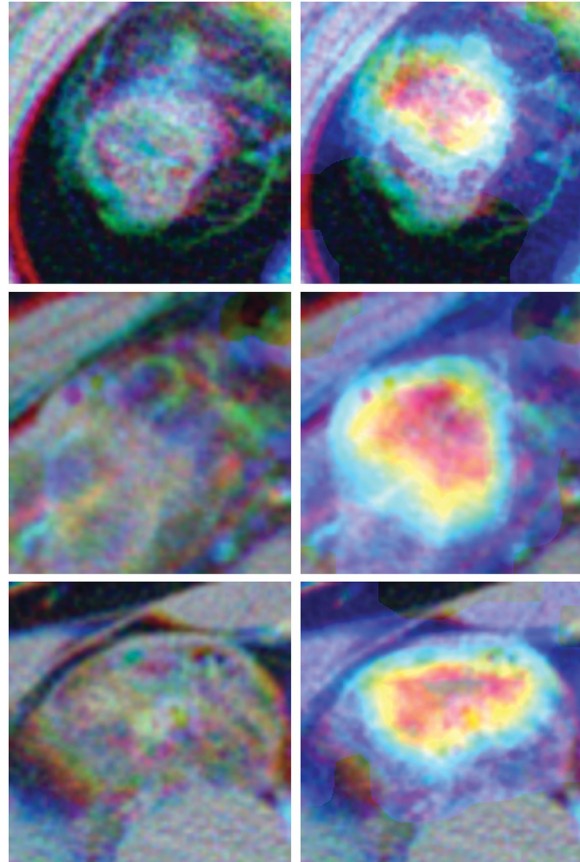

**Fig 6.** Examples of acute diverticulitis cases correctly classified by the convoluted neural network (CNN) using the (a) single image and (b) red, green, blue (RGB) methods, with and without class activation maps (CAMs).

enhanced CT scans. They used eight CT slices from eight sequential CT images, whereas we used three serial CT images with the RGB superposition method, including the information from continuous CT images (i.e., previous, current [lesion], and next image). This allowed us to establish the three-dimensional volume of the lesion using the CNN. Our CNN maximizes the diagnostic performance of image classification by leveraging deep learning and maintains a low error rate while fine-tuning the hyperparameters from small available datasets [29]. Moreover, it can adequately handle accuracy saturation with increasing network depth [29].

The EfficientNet algorithm models can scale the depth, width, and resolution dimensions of a CNN [17]. Transfer learning using the ImageNet dataset can save time and computing resources [17]. The EfficientNet algorithm provides superior efficiency and higher accuracy than other CNNs, including GoogleNet, AlexNet, and MobileNetV2 [17]. In this study, we trained our CNN model using EfficientNet-B0 architecture to classify acute appendicitis, acute diverticulitis, and normal appendix. The model was determined by considering the image resolution of the original image (before image resizing), while the training time affected the number of parameters [30].

We hypothesized that the use of three serial CT images in the RGB method would improve the CNN performance for classifying the three output labels (acute appendicitis, acute diverticulitis, and normal appendix) by enhancing the spatial features of the target lesion. Few studies

## - (a) Single methods -   - (b) RGB methods -

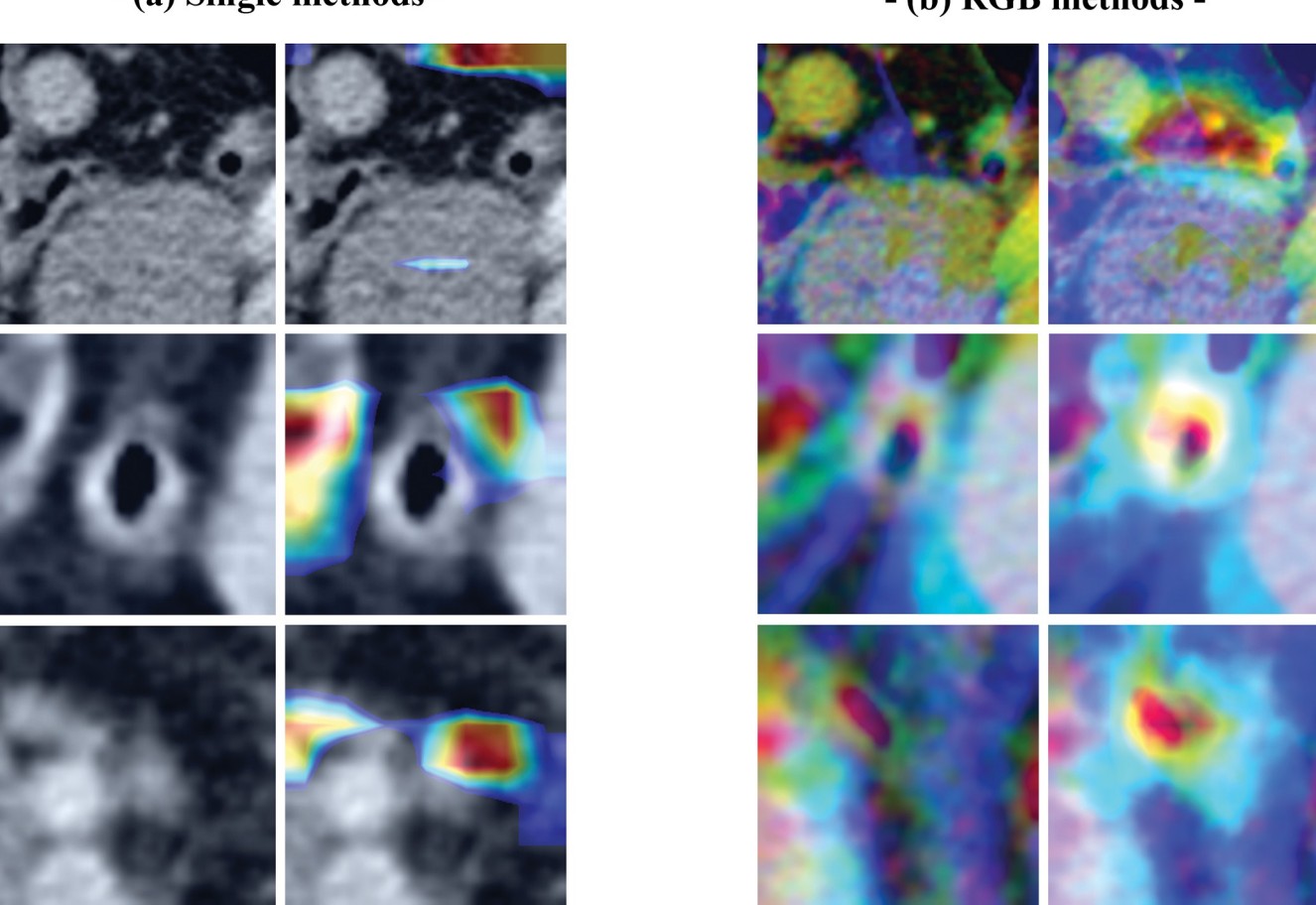

**Fig 7.** Examples of normal appendix cases correctly classified by the convoluted neural network (CNN) using the (a) single image and (b) red, green, blue (RGB) methods, with and without class activation maps (CAMs).

have reported CNNs using three CT slices with the RGB method [15, 16, 31] or multiple CT slices from videos [28, 32, 33] to enhance spatiotemporal features. The sensitivity of the model for classifying acute diverticulitis was slightly higher when using the RGB method (83.35%) than when using the single image method (80.44%), as was the specificity (96.04% and 95.12%, respectively). The sensitivity and specificity for classifying acute appendicitis were also slightly higher when using the RGB method (87.85% and 88.01%, respectively) than when using the single image method (85.60% and 86.50, respectively). The AUCs obtained with the RGB method (0.951 for acute appendicitis; 0.972 for acute diverticulitis; and 0.979 for normal appendix) were significantly higher than those obtained with the single image method (0.937, 0.963, and 0.972, respectively; $P < 0.01$). Therefore, spatial information from multiple slices may contribute to the classification accuracy when using contrast-enhanced abdominopelvic CT scans.

There are some limitations of our study that must be addressed. First, because this was a retrospective study, a selection bias may have occurred. Thus, a prospective study should be conducted to validate our classification results. Second, the number of CT images diagnosed as acute diverticulitis was relatively smaller than those diagnosed as acute appendicitis or normal appendix. Therefore, we analyzed the training dataset after augmenting each CT dataset to

minimize the training disturbance caused by unbalanced datasets. Third, we did not include cases of acute diverticulitis with complications (e.g., perforation or abscess) as such cases were beyond the scope of our study but will be addressed in our future studies. Fourth, we did not develop a localization tool to detect normal appendix, appendicitis, or acute diverticulitis. We therefore believe that identifying the location of such conditions using artificial intelligence will be helpful [11]. The development of such detection tools will be attempted in a future study. Finally, we did not assess the CNN performance using coronal reformatted CT images. Notably, combining axial and coronal CT images may enhance CNN performance, and we plan to investigate this combination in the future.

In conclusion, we developed an automated CNN-based model employing EfficientNet for classifying acute appendicitis, acute diverticulitis, and normal appendix on contrast-enhanced CT scans. All the aforementioned conditions were better classified by the CNN when serial images were used with the RGB method than when a single image method was used. This presumably reflects the detailed and rich spatial information provided by the former, which leverages information from multiple CT images.

## Supporting information

**S1 Checklist. STROBE statement.**
(DOC)

**S1 Appendix. Five-fold cross-validation.** (a,b) Single methods (c,d) RGB methods for classification of acute appendicitis, acute diverticulitis, and normal appendix using EfficientNet.
(DOCX)

## Author Contributions

**Conceptualization:** Jun-Won Chung.

**Data curation:** So Hyun Park, Young Jae Kim.

**Formal analysis:** Young Jae Kim, In Young Choi, Myung-Won You, Gi Pyo Lee.

**Funding acquisition:** Kwang Gi Kim.

**Investigation:** Gi Pyo Lee.

**Methodology:** Kwang Gi Kim, Hyun Cheol Kim.

**Writing – original draft:** So Hyun Park, Gi Pyo Lee.

**Writing – review & editing:** Kwang Gi Kim, Hyun Cheol Kim, In Young Choi, Myung-Won You, Jung Han Hwang.

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
