## [Decision Letter · Decision Letter 0]

16 Aug 2022

PONE-D-22-20125Comparison between single and serial computed tomography images in classification among acute appendicitis, acute diverticulitis, and normal appendix using EfficientNetPLOS ONE

Dear Dr. Kim,

Thank you for submitting your manuscript to PLOS ONE. After careful consideration, we feel that it has merit but does not fully meet PLOS ONE’s publication criteria as it currently stands. Therefore, we invite you to submit a revised version of the manuscript that addresses the points raised during the review process.

ACADEMIC EDITOR: Please revise the manuscript based on the reviewers suggestions and comments. 

We look forward to receiving your revised manuscript.

Kind regards,

Kathiravan Srinivasan

Academic Editor

PLOS ONE

Journal Requirements:

"This research was supported by the Ministry of Science and ICT (MSIT), Korea, under the Infor-mation Technology Research Center (ITRC) support program (IITP−2021−2017−0−01630) super-vised by the Institute for Information and communications Technology Promotion (IITP), and the research was supported by G−ABC FRD2019−11−02(3)"

"This research was supported by the Ministry of Science and ICT (MSIT), Korea, under the Infor-mation Technology Research Center (ITRC) support program (IITP−2021−2017−0−01630) super-vised by the Institute for Information and communications Technology Promotion (IITP), and the research was supported by G−ABC FRD2019−11−02(3)."

6. Please ensure that you refer to Figure 1 in your text as, if accepted, production will need this reference to link the reader to the figure.

Reviewers' comments:

Reviewer's Responses to Questions

**Comments to the Author**

1. Is the manuscript technically sound, and do the data support the conclusions?

Reviewer #1: Yes

Reviewer #2: Yes

Reviewer #3: Yes

Reviewer #4: Yes

2. Has the statistical analysis been performed appropriately and rigorously? 

Reviewer #1: Yes

Reviewer #2: Yes

Reviewer #3: Yes

Reviewer #4: Yes

3. Have the authors made all data underlying the findings in their manuscript fully available?

Reviewer #1: Yes

Reviewer #2: No

Reviewer #3: No

Reviewer #4: No

4. Is the manuscript presented in an intelligible fashion and written in standard English?

Reviewer #1: Yes

Reviewer #2: Yes

Reviewer #3: Yes

Reviewer #4: Yes

5. Review Comments to the Author

Reviewer #1: --I would like to congratulate the authors on their work.

--I have several questions and suggestions after reviewing the manuscript.

Title: Comparison between single and serial computed tomography images in classification among acute appendicitis, acute diverticulitis, and normal appendix using EfficientNet

--I think "right-sided" diverticulitis should be added to the title.

Introduction:

From the text: Although clinical presentations of two diseases are similar, the treatments differ between surgical and medical treatment [6-9].

--This is not true and should be stated as right-sided.

--Clinical presentations of acute appendicitis (AA) and acute right-sided colonic diverticulitis (ARCD) are similar. (Song JH, Kim YW, Lee S, Do HH, Seo JS, Lee JH, Lee SC. Clinical Difference between Acute Appendicitis and Acute Right-Sided Colonic Diverticulitis. Emerg Med Int. 2020 Sep 1;2020:4947192. doi: 10.1155/2020/4947192. PMID: 32953179; PMCID: PMC7481953.)

--The left-sided predominance of pain is due to the fact that most diverticulitis occurs in the sigmoid or descending colon. However, in Asian populations, diverticulitis is predominantly right-sided, and therefore pain is more often located on the right. (Swanson SM, Strate LL. Acute Colonic Diverticulitis. Ann Intern Med. 2018 May 1;168(9):ITC65-ITC80. doi: 10.7326/AITC201805010. Erratum in: Ann Intern Med. 2020 May 5;172(9):640. PMID: 29710265; PMCID: PMC6430566.)

From the text: Acute appendicitis requiring surgery [10] and acute diverticulitis requiring medical treatment in uncomplicated cases [11].

--This information is incorrect; some references (Addiss DG, Shaffer N, Fowler BS, Tauxe RV. The epidemiology of appendicitis and appendectomy in the United States. American journal of epidemiology. 1990;132(5): 910- 925.) are outdated.

--References should be selected from up-to-date publications. ( Ergenç M, Uprak TK. Appendiceal Diverticulitis Presenting as Acute Appendicitis and Diagnosed After Appendectomy. Cureus. 2022 Mar 10;14(3). doi:10.7759/cureus.23050 )

Methods:

From the text: For the acute appendicitis group, the inclusion criteria were as follows: (a) patients (≥ 20 year old) with acute lower abdominal pain showed acute appendicitis detected via CT (i.e., we searched CT reports to find acute appendicitis and reviewed the clinical symptoms) and surgically confirmed and (b) preoperative abdominopelvic CT within 1 week of surgery.

--Were patients with abscesses or appendix perforation excluded?

From the text: For the acute diverticulitis group, the inclusion criteria were as follows: (a) patients (≥ 20 year old) with acute lower abdominal pain and acute diverticulitis detected via CT and (b) medically treated cases that improved during follow-up (>1 month). The exclusion criteria were (a) bowel perforation due to acute diverticulitis and (b) severe motion artifacts on CT images.

--Please give localization of diverticulitis. (Right side, sigmoid or descending)

From the text: We manually cropped box-shaped region of interests (ROIs) at the same position across all axial planes from the start to the end positions of the visible targets for acute appendicitis, acute diverticulitis, and normal appendix. Appropriate classification of a normal appendix using the CNN model can help inexperienced physicians who have to decide whether the appendix is in a normal state on CT in the emergency room.

--It will be more helpful if artificial intelligence makes this localization. (Al WA, Yun ID, Lee KJ. Reinforcement Learning-based Automatic Diagnosis of Acute Appendicitis in Abdominal CT. arXiv preprint arXiv:190900617. 2019.)

Result:

From the text: The test dataset contained 4078 CT image slices (acute appendicitis, 1959; acute diverticulitis, 823; normal appendix, 1296), which were 189 derived from scans of 715 patients.

--Is it necessary to have radiology training to select these CT slices? How will "inexperienced physicians" do this?

Discussion:

From the text: We evaluated 715 patients who underwent abdominopelvic CT and were diagnosed with acute appendicitis, acute diverticulitis, or normal appendix. We developed and applied a CNN model using the EfficientNetB0 architecture to classify acute appendicitis, acute diverticulitis, and normal appendix on the single and RGB contrast-enhanced abdominopelvic CT images. The RGB method showed an AUC of 0.951 for acute appendicitis, 0.972 for acute diverticulitis, and 0.979 for normal appendix; all AUC values in the RGB methods were higher than those in the single method (0.937, 0.963, and 0.972; P <0.01).

--This section is suitable for "results", and should not be in "discussion".

From the text: Third, we did not include cases of acute diverticulitis with complications (e.g., perforation or abscess).

--Complicated cases of appendicitis were not mentioned in the limitation section.

Reviewer #2: Literature Survey is inadequate. More no of papers need to cite ,

Introduction about image processing need to be done.

Methodology to be more clear. In detailed explanation needed.

comparison aspects need to improve.

Reviewer #3: The Statistical Analysis and comparison of images can be elaborated and the results could be improved and can be justified in the conclusion. The Comparison between images could be expanded with much better images for the result to be accurate. the images taken authenticity is a big question, hope that can be addressed.

Reviewer #4: In this article, the authors have employed EfficientNet to classify acute appendicitis, acute diverticulitis, and normal appendix.

Although the manuscript is technically sound and the results seem correct, there is no novelty in the method or the model that has been used.

I have some comments/questions about this study:

1- The quality of the figures and the flowchart are very low.

2- In applying deep learning methods to medical images, it is important to provide evidence on how the model decides about the class of the images. There are some interpretability methods that can be used for this purpose. I would like to see some interpretability analysis of the resulted model.

3- Please explain more about the EfficientNet and why it was chosen as the CNN architecture. There are some other CNNs with better accuracy on benchmark datasets.

6. PLOS authors have the option to publish the peer review history of their article (what does this mean?). If published, this will include your full peer review and any attached files.

Reviewer #1: No

Reviewer #2: **Yes: **Mudrakola Swapna

Reviewer #3: **Yes: **Dr.P.Kavitha Rani

Reviewer #4: No

---

## [Author Response · Author response to Decision Letter 0]

2 Oct 2022

Response to reviewer

Reviewers' comments:

Reviewer's Responses to Questions

Comments to the Author

1. Is the manuscript technically sound, and do the data support the conclusions?

Reviewer #1: Yes

Reviewer #2: Yes

Reviewer #3: Yes

Reviewer #4: Yes

2. Has the statistical analysis been performed appropriately and rigorously?

Reviewer #1: Yes

Reviewer #2: Yes

Reviewer #3: Yes

Reviewer #4: Yes

3. Have the authors made all data underlying the findings in their manuscript fully available?

Reviewer #1: Yes

Reviewer #2: No

Reviewer #3: No

Reviewer #4: No

4. Is the manuscript presented in an intelligible fashion and written in standard English?

Reviewer #1: Yes

Reviewer #2: Yes

Reviewer #3: Yes

Reviewer #4: Yes

5. Review Comments to the Author

Reviewer #1: --I would like to congratulate the authors on their work.

--I have several questions and suggestions after reviewing the manuscript.

Title: Comparison between single and serial computed tomography images in classification among acute appendicitis, acute diverticulitis, and normal appendix using EfficientNet

--I think "right-sided" diverticulitis should be added to the title.

Response) Thank you. We corrected the title as your suggestion (R1-1).

Introduction:

From the text: Although clinical presentations of two diseases are similar, the treatments differ between surgical and medical treatment [6-9].

--This is not true and should be stated as right-sided.

--Clinical presentations of acute appendicitis (AA) and acute right-sided colonic diverticulitis (ARCD) are similar. (Song JH, Kim YW, Lee S, Do HH, Seo JS, Lee JH, Lee SC. Clinical Difference between Acute Appendicitis and Acute Right-Sided Colonic Diverticulitis. Emerg Med Int. 2020 Sep 1;2020:4947192. doi: 10.1155/2020/4947192. PMID: 32953179; PMCID: PMC7481953.)

--The left-sided predominance of pain is due to the fact that most diverticulitis occurs in the sigmoid or descending colon. However, in Asian populations, diverticulitis is predominantly right-sided, and therefore pain is more often located on the right. (Swanson SM, Strate LL. Acute Colonic Diverticulitis. Ann Intern Med. 2018 May 1;168(9):ITC65-ITC80. doi: 10.7326/AITC201805010. Erratum in: Ann Intern Med. 2020 May 5;172(9):640. PMID: 29710265; PMCID: PMC6430566.)

From the text: Acute appendicitis requiring surgery [10] and acute diverticulitis requiring medical treatment in uncomplicated cases [11].

--This information is incorrect; some references (Addiss DG, Shaffer N, Fowler BS, Tauxe RV. The epidemiology of appendicitis and appendectomy in the United States. American journal of epidemiology. 1990;132(5): 910- 925.) are outdated.

--References should be selected from up-to-date publications. ( Ergenç M, Uprak TK. Appendiceal Diverticulitis Presenting as Acute Appendicitis and Diagnosed After Appendectomy. Cureus. 2022 Mar 10;14(3). doi:10.7759/cureus.23050 )

Response) Thank you. We corrected the references as your suggestion (R1-2) and added "right-sided" to describe acute diverticulitis (R1-2) in the introduction.

Methods:

From the text: For the acute appendicitis group, the inclusion criteria were as follows: (a) patients (≥ 20 year old) with acute lower abdominal pain showed acute appendicitis detected via CT (i.e., we searched CT reports to find acute appendicitis and reviewed the clinical symptoms) and surgically confirmed and (b) preoperative abdominopelvic CT within 1 week of surgery.

--Were patients with abscesses or appendix perforation excluded?

Response) The patients with abscesses or appendix perforation were not excluded. However, a patient who underwent preoperative abdominopelvic CT more than 1 week of surgery were not included. Thus, some cases with periappendiceal abscess which performed surgery after 1 week of CT scan were not included.

From the text: For the acute diverticulitis group, the inclusion criteria were as follows: (a) patients (≥ 20 year old) with acute lower abdominal pain and acute diverticulitis detected via CT and (b) medically treated cases that improved during follow-up (>1 month). The exclusion criteria were (a) bowel perforation due to acute diverticulitis and (b) severe motion artifacts on CT images.

--Please give localization of diverticulitis. (Right side, sigmoid or descending)

Response) Thank you. We added the site of acute diverticulitis: "right-sided" (R1-3).

From the text: We manually cropped box-shaped region of interests (ROIs) at the same position across all axial planes from the start to the end positions of the visible targets for acute appendicitis, acute diverticulitis, and normal appendix. Appropriate classification of a normal appendix using the CNN model can help inexperienced physicians who have to decide whether the appendix is in a normal state on CT in the emergency room.

--It will be more helpful if artificial intelligence makes this localization. (Al WA, Yun ID, Lee KJ. Reinforcement Learning-based Automatic Diagnosis of Acute Appendicitis in Abdominal CT. arXiv preprint arXiv:190900617. 2019.)

Response) Thank you. We added the sentence according your suggestion in the limitation (R1-4). "We also believe that it will be more helpful if artificial intelligence can find the localization."

Result:

From the text: The test dataset contained 4078 CT image slices (acute appendicitis, 1959; acute diverticulitis, 823; normal appendix, 1296), which were 189 derived from scans of 715 patients.

--Is it necessary to have radiology training to select these CT slices? How will "inexperienced physicians" do this?

Response) Our study only analyzed the lesion site to classify three states (appendix, acute appendicitis, and diverticulitis) when a physician found a suspicious site. Future study will explore detection tools. We added the limitation (R1-5). 

Discussion:

From the text: We evaluated 715 patients who underwent abdominopelvic CT and were diagnosed with acute appendicitis, acute diverticulitis, or normal appendix. We developed and applied a CNN model using the EfficientNetB0 architecture to classify acute appendicitis, acute diverticulitis, and normal appendix on the single and RGB contrast-enhanced abdominopelvic CT images. The RGB method showed an AUC of 0.951 for acute appendicitis, 0.972 for acute diverticulitis, and 0.979 for normal appendix; all AUC values in the RGB methods were higher than those in the single method (0.937, 0.963, and 0.972; P <0.01).

--This section is suitable for "results", and should not be in "discussion".

Response) We deleted several sentences as your suggestion (R1-6) in discussion.

From the text: Third, we did not include cases of acute diverticulitis with complications (e.g., perforation or abscess).

--Complicated cases of appendicitis were not mentioned in the limitation section.

Response) The patients with abscesses or appendix perforation were not excluded. However, a patient who underwent preoperative abdominopelvic CT more than 1 week of surgery were not included. 

Our target population was surgically confirmed acute diverticulitis and medically treated acute diverticulitis in patients who underwent CT scan within 1 week.

Reviewer #2: Literature Survey is inadequate. More no of papers need to cite ,

Introduction about image processing need to be done.

Methodology to be more clear. In detailed explanation needed.

comparison aspects need to improve.

Response) We added several references as your suggestion (R2-1). We added image procession explanation in the introduction section and Implementation and training of the deep learning model section (R2-1); The EfficientNet algorithm is composed of 8 models from B0 to B7 and the higher number refers to more parameters and higher accuracy. It influences neural architecture search to explore the baseline EfficientNet-B0 that has better trade-off on accuracy and calculation time (i.e. floating-point operations per second). We have used the B0 model which covers 4 million trainable parameters.; In applying deep learning model, the result showed the prediction probability of three statuses. Among the results, the highest probability for the three revealed the outcome of the prediction label name (Fig 3b).

Reviewer #3: The Statistical Analysis and comparison of images can be elaborated and the results could be improved and can be justified in the conclusion. The Comparison between images could be expanded with much better images for the result to be accurate. the images taken authenticity is a big question, hope that can be addressed.

Response) Thank you for your comment. We added the model explanation (R3-1) and changed the figures to improve image quality and to explain the process well (Fig 3 a-b). In applying deep learning model, the result showed the prediction probability for three statuses. Among the results, the highest probability for the three revealed the outcome of the prediction label name (Fig 3b). And we added the limitation in our study; we did not develop a localization tool to detect appendix, appendicitis, or acute diverticulitis (R3-1).

Reviewer #4: In this article, the authors have employed EfficientNet to classify acute appendicitis, acute diverticulitis, and normal appendix.

Although the manuscript is technically sound and the results seem correct, there is no novelty in the method or the model that has been used.

I have some comments/questions about this study:

1- The quality of the figures and the flowchart are very low.

Response) Thank you for your comment. We changed the figures to improve image quality and to explain the process well (R4-1).

2- In applying deep learning methods to medical images, it is important to provide evidence on how the model decides about the class of the images. There are some interpretability methods that can be used for this purpose. I would like to see some interpretability analysis of the resulted model.

Response) Thank you for your comment. We added the model explanation (R4-2); In applying deep learning model, the result showed the prediction probability of three statuses [probability for acute appendicitis, acute diverticulitis, normal appendix]. Among the results, the highest probability among the three revealed the outcome of the prediction label name (Fig 3b).

3- Please explain more about the EfficientNet and why it was chosen as the CNN architecture. There are some other CNNs with better accuracy on benchmark datasets.

Response) We initially analyzed the results with Resnet a few years ago. We found that several studies developed deep learning-medical image classification using EfficientNet (1. references, 2. introduction of EfficientNet). Thus, we tried to compare the results between EffectintNet B0 (which showed similar performance to Resnet 50) and Resnet 50. However, the result using EffectintNet showed much more excellent performance than that using Resnet 50 and the result using B0 was sufficiently good at classifying lesions, and we analyzed the study using EffectintNet-B0 model only. Currently, other CNNs show better performance and the technology is advancing very rapidly. We planned a future study to compare performances among EfficientNetB0-B7 for classification.

1) (References.1. Marques G, Agarwal D, de la Torre Díez I. Automated medical diagnosis of COVID-19 through EfficientNet convolutional neural network. Appl Soft Comput. 2020;96: 106691-106691. doi: 10.1016/j.asoc.2020.106691.

2. Duong LT, Nguyen PT, Di Sipio C, Di Ruscio D. Automated fruit recognition using EfficientNet and MixNet. Computers and Electronics in Agriculture. 2020;171: 105326. doi: https://doi.org/10.1016/j.compag.2020.105326.

3. Alhichri H, Alswayed AS, Bazi Y, Ammour N, Alajlan NA. Classification of Remote Sensing Images Using EfficientNet-B3 CNN Model With Attention. IEEE Access. 2021;9: 14078-14094. doi: 10.1109/ACCESS.2021.3051085.

4. Mantha T, Reddy BE, editors. A Transfer Learning method for Brain Tumor Classification using EfficientNet-B3 model. 2021 IEEE International Conference on Computation System and Information Technology for Sustainable Solutions (CSITSS); 2021 16-18 Dec. 2021.)

2) Introduction: The EfficientNet algorithm is an introduced method that uses compounded coefficients to scale the depth, width, and resolution dimensions of a CNN simultaneously and eventually we can obtain a better performance with higher accuracy and shorter time using RGB superposition. It weights neural architecture search to find for the baseline EfficientNet-B0 that has better trade-off on accuracy and calculation time (i.e. floating-point operations per second).

6. PLOS authors have the option to publish the peer review history of their article (what does this mean?). If published, this will include your full peer review and any attached files.

Do you want your identity to be public for this peer review? For information about this choice, including consent withdrawal, please see our Privacy Policy.

Reviewer #1: No

Reviewer #2: Yes: Mudrakola Swapna

Reviewer #3: Yes: Dr.P.Kavitha Rani

Reviewer #4: No

---

## [Decision Letter · Decision Letter 1]

26 Oct 2022

PONE-D-22-20125R1Comparison between single and serial computed tomography images in classification among acute appendicitis, acute right-sided diverticulitis, and normal appendix using EfficientNetPLOS ONE

Dear Dr. Kim,

Thank you for submitting your manuscript to PLOS ONE. After careful consideration, we feel that it has merit but does not fully meet PLOS ONE’s publication criteria as it currently stands. Therefore, we invite you to submit a revised version of the manuscript that addresses the points raised during the review process.

ACADEMIC EDITOR: Kindly revise the manuscript as suggested by the reviewers.

Also, consider improving the English language presentation.

We look forward to receiving your revised manuscript.

Kind regards,

Kathiravan Srinivasan

Academic Editor

PLOS ONE

Journal Requirements:

Reviewers' comments:

Reviewer's Responses to Questions

**Comments to the Author**

1. If the authors have adequately addressed your comments raised in a previous round of review and you feel that this manuscript is now acceptable for publication, you may indicate that here to bypass the “Comments to the Author” section, enter your conflict of interest statement in the “Confidential to Editor” section, and submit your "Accept" recommendation.

Reviewer #1: (No Response)

Reviewer #2: All comments have been addressed

Reviewer #3: All comments have been addressed

2. Is the manuscript technically sound, and do the data support the conclusions?

Reviewer #1: Yes

Reviewer #2: Yes

Reviewer #3: Yes

3. Has the statistical analysis been performed appropriately and rigorously? 

Reviewer #1: Yes

Reviewer #2: Yes

Reviewer #3: Yes

4. Have the authors made all data underlying the findings in their manuscript fully available?

Reviewer #1: No

Reviewer #2: Yes

Reviewer #3: Yes

5. Is the manuscript presented in an intelligible fashion and written in standard English?

Reviewer #1: Yes

Reviewer #2: No

Reviewer #3: Yes

6. Review Comments to the Author

Reviewer #1: After my and other reviewers' comments, I thank the authors for their responses and changes to the article. After these changes and additions, I think the article has improved and is suitable for publication. However, some references are outdated (from 1990 and 1998), and I recommend using current references instead.

Reviewer #2: All review comment of my earlier suggestion are been addressed by the author, Author improvised the paper as per the reviewer comment

Reviewer #3: All the comments that was addressed in the earlier review is updated in the article, it can be published in the journal

7. PLOS authors have the option to publish the peer review history of their article (what does this mean?). If published, this will include your full peer review and any attached files.

Reviewer #1: No

Reviewer #2: **Yes: **Swapna Mudrakola

Reviewer #3: No

---

## [Author Response · Author response to Decision Letter 1]

6 Jan 2023

Reviewer #1: After my and other reviewers' comments, I thank the authors for their responses and changes to the article. After these changes and additions, I think the article has improved and is suitable for publication. However, some references are outdated (from 1990 and 1998), and I recommend using current references instead.

Response) We have corrected several references as your suggestion.

Reviewer #2: All review comment of my earlier suggestion are been addressed by the author, Author improvised the paper as per the reviewer comment

Response) Thank you. And we corrected grammatical errors and tried some attention to awkwardly phrased sentences.

Reviewer #3: All the comments that was addressed in the earlier review is updated in the article, it can be published in the journal

Response) Thank you.

---

## [Decision Letter · Decision Letter 2]

25 Jan 2023

Comparison between single and serial computed tomography images in classification of acute appendicitis, acute right-sided diverticulitis, and normal appendix using EfficientNet

PONE-D-22-20125R2

Dear Dr. Kim,

We’re pleased to inform you that your manuscript has been judged scientifically suitable for publication and will be formally accepted for publication once it meets all outstanding technical requirements.

Kind regards,

Kathiravan Srinivasan

Academic Editor

PLOS ONE

Additional Editor Comments (optional):

Reviewers' comments:

Reviewer's Responses to Questions

**Comments to the Author**

1. If the authors have adequately addressed your comments raised in a previous round of review and you feel that this manuscript is now acceptable for publication, you may indicate that here to bypass the “Comments to the Author” section, enter your conflict of interest statement in the “Confidential to Editor” section, and submit your "Accept" recommendation.

Reviewer #1: All comments have been addressed

Reviewer #2: All comments have been addressed

Reviewer #3: All comments have been addressed

2. Is the manuscript technically sound, and do the data support the conclusions?

Reviewer #1: Yes

Reviewer #2: Yes

Reviewer #3: Partly

3. Has the statistical analysis been performed appropriately and rigorously? 

Reviewer #1: Yes

Reviewer #2: Yes

Reviewer #3: Yes

4. Have the authors made all data underlying the findings in their manuscript fully available?

Reviewer #1: Yes

Reviewer #2: Yes

Reviewer #3: Yes

5. Is the manuscript presented in an intelligible fashion and written in standard English?

Reviewer #1: Yes

Reviewer #2: Yes

Reviewer #3: Yes

6. Review Comments to the Author

Reviewer #1: After my and other reviewers' comments, I thank the authors for their responses and changes to the article. After these changes and additions, I think the article has improved and is suitable for publication.

Reviewer #2: (No Response)

Reviewer #3: authors have addressed the comments & all data underlying the findings described in their manuscript the authors can improvise the technical background in their future manuscript which will help the research work

7. PLOS authors have the option to publish the peer review history of their article (what does this mean?). If published, this will include your full peer review and any attached files.

Reviewer #1: No

Reviewer #2: **Yes: **Mudrakola Swapna

Reviewer #3: No

---

## [Editor Report · Acceptance letter]

16 May 2023

PONE-D-22-20125R2 

Comparison between single and serial computed tomography images in classification of acute appendicitis, acute right-sided diverticulitis, and normal appendix using EfficientNet 

Dear Dr. Kim:

I'm pleased to inform you that your manuscript has been deemed suitable for publication in PLOS ONE. Congratulations! Your manuscript is now with our production department. 

Kind regards, 

on behalf of

Dr. Kathiravan Srinivasan 

Academic Editor

PLOS ONE